# Detecting Topological Defects in 2D Active Nematics Using Convolutional Neural Networks

## Abstract

Active matter consists of active agents which transform energy extracted from surroundings into momentum, producing a variety of collective phenomena. A model, synthetic active system composed of microtubule polymers driven by protein motors spontaneously forms a liquid-crystalline nematic phase. Extensile stress created by the protein motors precipitates continuous buckling and folding of the microtubules creating motile topological defects and turbulent fluid flows. Defect motion is determined by the rheological properties of the material; however, these remain largely unquantified. Measuring defects dynamics can yield fundamental insights into active nematics, a class of materials that include bacterial films and animal cells. Current methods for defect detection lack robustness and precision, and require fine-tuning for datasets with different visual quality. In this study, we applied Deep Learning to train a defect detector to automatically analyze microscopy videos of the microtubule active nematic. Experimental results indicate that our method is robust and accurate. It is expected to significantly increase the amount of video data that can be processed.

## 1 Introduction

### 1.1 Active Matter and Active Nematics

Active materials encompass a broad range of systems that convert chemical energy into mechanical work. These systems self-organize and exhibit structure on time and length scales that exceed that of the constituent particles. One important class of active materials is active-nematics composed of anisotropic particles that exert extensile or contractile stresses on neighboring particles. Natural examples of active nematics include cultures of dividing *E. coli* and animal cells Doostmohammadi et al. (2016); Segerer et al. (2015); Duclos et al. (2014; 2017). In active nematics, a single particle does not impart a net force into the system and therefore does self-propel; however, the collective motion of many particles create non-trivial material flows, Marchetti et al. (2013). Understanding the fundamental physics of collective motion of nematics can yield insight wound healing, tumor growth, and bacterial film dynamics, Doostmohammadi et al. (2018).

### 1.2 2D Confined Active Nematics and Topological Defects

In this work we study a model quasi-2D active nematic system composed of microtubules (MT) and motor proteins with tunable mechanical and dynamical properties developed by the Dogic group Henkin et al. (2014); DeCamp et al. (2015). In this system, a suspension of microtubules and motor proteins is sedimented to a surfactant-stabilized oil-water interface creating a dense, liquid-crystalline nematic phase characterized by local orientational order. Motor proteins drive neighboring MTs to slide anti-parallel to one another, creating extensile stresses. The extensile stress makes the MT nematic inherently unstable to bend fluctuations. Undulations in the alignment of the MTs therefore grow in time, eventually saturating into localized topological disclinations shown in figure 1. The defects are characterized by their winding number Kamien (2002). Unlike defects in passive liquid crystals, these defects create fluid flows. The comet-shaped +1/2 defects are self-propelling while the three-fold symmetry of the -1/2 defects create active flows that result in to net translation of the structure; active flows are shown by the yellow arrows in figure 1.

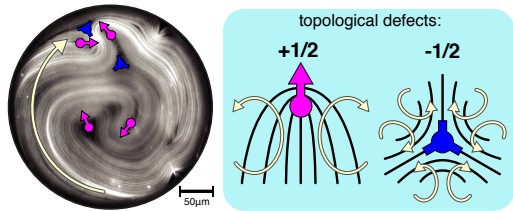

Figure 1: **(Left)** Fluorescently labeled active microtubule suspension confined to a 300 $\mu$m diameter well. $+\frac{1}{2}$ and $-\frac{1}{2}$ defects are labeled as magenta arrows and blue dots with three prongs, respectively. Yellow arrow indicates the direction of net circulation in the sample. **(Right)** Schematic representation of topological defects. Black lines indicate the director field, yellow arrows indicate the active flow created by the collective motion of microtubules Doostmohammadi et al. (2018).

The hydrodynamic interactions between multiple defects create chaotic, turbulent-like flows. By further confining the 2D nematic to microfluidic wells, these flows can be tamed into circulating vortices Woodhouse & Goldstein (2012); Norton et al. (2018). Defect dynamics play a key role in moderating the transition from bulk-like turbulence to regular circulation. Quantifying their dynamics will help develop a better understanding of the fluid dynamics governing the behavior of this complex fluid and perhaps give insights into generic features of motile defects in biological systems. This requires researchers to analyze a large volume of videos of the 2D confined active nematics.

## 1.3 TRADITIONAL DEFECT DETECTING ALGORITHM AND LIMITATIONS

Remarkably utilizing the defining feature of a topological defect – its winding number, researchers developed an image processing algorithm which first generates a direction field from each image and then calculate the winding number at regions across the whole image to search for singularities. This algorithm performs reasonably well in detecting both +1/2 and -1/2 defects while being limited by two factors: various visual quality and noises caused by imperfect experimental settings. Parameters in the algorithm need to be fine-tuned for datasets with different visual qualities which would require huge amount of human labor as well as knowledge of image processing. The traditional algorithm also suffers from noises in the data including overexposure in some region, slightly unfocused image, materials exceeding above the 2D confinement plane, etc., which tend to ruin the direction fields extracted from the images resulting poor detecting results. Aside from precision, this algorithm involves intensive calculation when processing one image. Fine-tuning of parameters and slow processing speed greatly limits the efficiency of data analysis in the experiments, while the algorithm's lack of robustness compromising the detecting results.

## 1.4 PROPOSAL OF SOLUTION

To address the problems existing in the traditional defect detecting algorithm, we propose to apply deep learning, specifically deep convolutional neural network to replace the above defect detecting algorithm that is based on traditional image processing techniques. Our tasks include:

- Label the positions of defects in representative experimental data.
- Train a YOLO network with bounding boxes generated according to the positions of the labelled defects.
- Test the performance of the trained defect detection model on experimental data with various visual qualities and sizes

The changing visual quality, which greatly challenges the the traditional algorithm, improves the performance of our method. Adding more defects in various lighting conditions and image resolutions improves both robustness and precision of our algorithm.

Apart from the visual quality, the diversity of the defects configuration causes troubles in traditional detecting algorithm. However, with defects in different shapes, sizes, configuration, and sometimes

occlusion, our algorithm is able to generalize its model better to fit in these various samples, compromising the effect of overfitting.

Another important advantage is our detection algorithms utilizes a unified, end-to-end neural network which enables efficient training and testing process.

## 2 PATTERNS IN DATASET

The labeled dataset consists of 9 typical videos from different experimental settings. In each experiment, active nematics are confined by a circular plate. Due to the constant input of energy, the active nematics continuously flows in a turbulent-like manner. Figure 2 and 3 shows a snapshot from each video. Specifically, the first testing dataset is from the same video as the first training dataset but are separated by 500 frames so that the first image of testing dataset has completely different appearance than the last image of the training dataset.

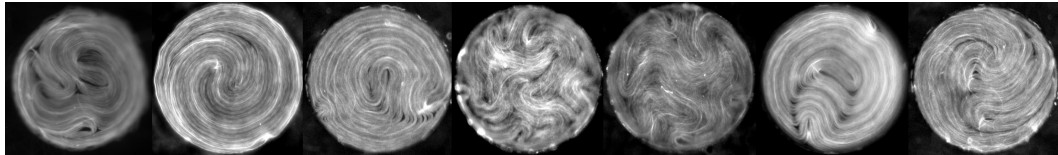

Figure 2: Snapshots from Each Training Dataset: fluorescently labelled microtubules in microfluidic wells ranging from 300-500$\mu$m in diameter. The first image is from a dataset of 6000 images and the rest each has 100 images.



Figure 3: Snapshots from Each Testing Dataset: fluorescently labelled microtubules in microfluidic wells ranging from 300-500$\mu$m in diameter. The first image is from a dataset of 1500 images and the rest each has 100 images.

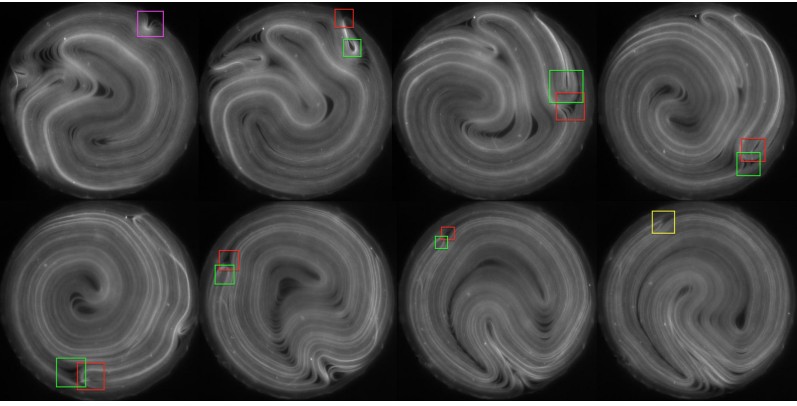

Figure 4: Life of a +1/2 Defect: from top-left to bottom-right shows the moving trajectory of a +1/2 defect from nucleation to annihilation. +1/2 defects are labeled by green boxes; -1/2 defects are labeled by red boxes; the nucleation process is labeled by magnum boxes; the annihilation process is labeled by yellow box. The diameter of the microfluidic wells in this figure is 300$\mu$m.

Figure 4 shows the moving trajectory of a typical +1/2 defect from nucleation to annihilation. In the first frame, a pair of +1/2 and -1/2 defects nucleates with +1/2 defect moving away from -1/2 defects. In the next six frames, the +1/2 defect rotates around the center while being pushed outwards by inner materials, encountering a -1/2 defects. In the last frame, the +1/2 and -1/2 defects annihilate concurrently, balancing the topological charge of the region.

## 3 METHODS

When dealing with defects detection, the most intuitive way is to find the characteristics of a defect which differ it from its neighborhood area and make use of these characteristics to locate the regions containing defects. The defining difference between a defect region and a non-defect one is the topological winding number. A region containing a defect has a winding number of +1/2 or -1/2 around its boundary while the winding number associated to a non-defect region is strictly 0. Previously, researchers made use of this definition and developed an effective algorithm by finding the direction field – the "grains" of an image, and calculating the winding number of different regions. As a result, this algorithm requires preprocessing to extract the direction field from the original image. Meanwhile, the performance of the detecting results relies heavily on the quality of the direction field extracted from the original image. Due to the imperfections and noises in the images, preprocessing is challenging. In some cases, the defects are occluded by overexposure area or are unfocused, leading to a problematic direction field near the defect. In addition, in the experimental video, materials such as hairs and dirts are observed which sometimes causes false detection.

To resolve these problems, we propose to treat this task as an object detection task where we consider defects as the objects we are trying to detect in an image. We train a convolutional neural network with images containing defects whose locations are provided. During detection, our model generates a bounding box around defects as detecting results. In this method, the problems in the previous paragraph, namely increasing the size of dataset from various visual quality, diversity of defects configuration and size, help our model generalize a defect's features better. Compared to the traditional algorithm which requires fine-tuning for different datasets, our method could efficiently provide reliable detecting results on data across various visual quality.

### 3.1 DATA COLLECTION

To create a training dataset, we manually labeled the positions of positive defects and negative defects in 8800 images. In an image, we labeled a comet-like (+1/2) defect at the point where materials change orientation most sharply. For a trefoil-like (-1/2) defect, we labeled them at the center of the triangular material-devoid region. Figure 5a is an example of labeled image with +1/2 defects labeled by red stars and -1/2 defects labeled by green circles.

### 3.2 SUB-CLASSIFICATION FOR +1/2 DEFECTS

The experimental datasets we work on are challenging to perform object detection. One main challenge is the varying sizes of +1/2 defects. In Figure 5b, there are 4 +1/2 defects by definition but defect 1 looks drastically different from defect 2, 3, and 4. We call +1/2 defects similar to defect 1 "hollow" +1/2 defects. Assigning bounding boxes with the same size for all four defects in this image is problematic. A small bounding box cannot include enough local information to recognize defect 1 as +1/2 defect, while big bounding boxes might include features of other defects which confuse the model. For defect 3 and 4, big bounding boxes will include the -1/2 defects next to them. To resolve this issue, since hollow +1/2 defects are commonly seen in the video, we created a separate class for these defects when we labeled the data and assigned slightly bigger bounding boxes for this class.

After labeling all the data, we assigned a bounding box to each labeled position. Since more contextual information is needed to detect hollow +1/2 defect than to detect normal +1/2 or -1/2 defects, we assigned the sizes of the bounding boxes with a ratio of 3:3:4 for +1/2, -1/2, hollow +1/2, respectively.

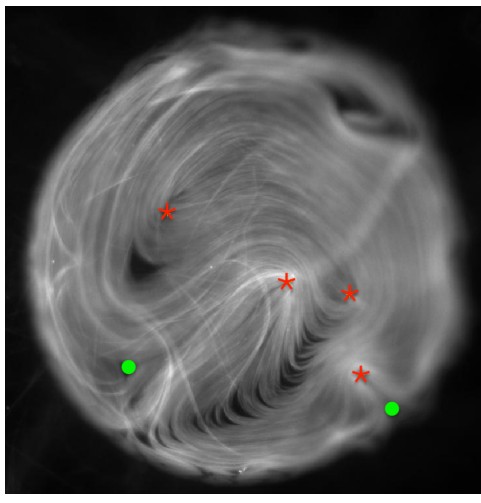 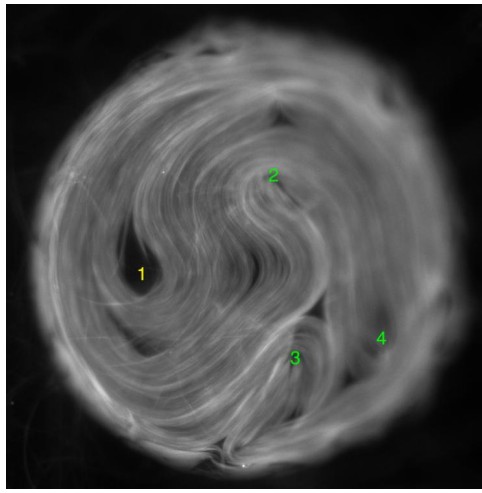

(a) Labeled Image  (b) +1/2 Defects

Figure 5: Experimental Data: fluorescently labelled microtubules in microfluidic wells with $400\mu$m in diameter.(**Left**: sample image with +1/2 and -1/2 defects labeled; **Right**: sample image containing +1/2 defects with 2 drastically different configurations and sizes.)

### 3.3 IMAGE AUGMENTATION

Aside from normal image augmentation such as jitter and scaling, we added a random flipping and random rotation to the training images. Since the region of a defect is anisotropic, rotating the image effectively increase the robustness of the model.

### 3.4 COMPUTER VISION PIPELINE

In recent years, after the wide application of convolutional neural network in computer vision, the state-of-the-art object detection algorithm can be divided into two main categories. In the first category, input images are first processed by a method named selective search Uijlings et al. (2013) which generates thousands of region proposals that are likely to contain an object. These region proposals along with the input images will be passed to a CNN which performs feature extractions and image re-sampling. The results will eventually be generated by a classifier. This category of algorithms is the first to combine a selective search method with deep CNN where selective search perform the task of object localization and CNN performs the task of classification. RCNN series (including R-CNN Girshick et al. (2014), Fast R-CNN Girshick (2015), and Faster R-CNN Ren et al. (2015)) belongs to this catogary. The second category is YOLO Redmon et al. (2016), which implements an end-to-end deep neural network that performs classification and localization all at once. The input images are first separated by 13 by 13 grid cells where each grid cell is responsible for detecting the object falling into it. The output of the network is forced to be a tensor which includes the coordinates of bounding boxes and a confidence value. YOLO outperforms its counterpart in speed but lack in average precision.

In this project, we chose to implement YOLO because it is fast and easy to optimize due to its end-to-end design. While implementing the YOLO algorithm, we mainly used the original setting in its paper except for some changes in the pipeline:

- Changed the number of filter in the last convolutional layer to fit in our dataset.

- Decreased the jitter value from 0.3 to 0.05 to avoid occlusion or missing of defects located near the boundary

- Since our objects-defects are relatively smaller than objects in the PASCAL VOC dataset, we increased the non-object coefficient from 0.5 to 0.8 to increase the loss from confidence predictions for boxes that do not contain objects

- Took out the image augmentation regarding hue, exposure, and saturation since our images are binary

## 4 RESULTS

### 4.1 DETECTING RESULTS COMPARISON

Table 1: Detecting Results of Traditional Image Processing Algorithm

| Traditional | Overall | Dataset 1 | Dataset 2 | Dataset 3 |
|---|---|---|---|---|
| Precision of $+1/2$ defects | 0 6156 | 0 6105 | 0 8796 | 0 5521 |
| Precision of $+1/2$ defects | 0 3649 | 0 3477 | 0 6022 | 0 3957 |
| Precision Overall | 0 5212 | 0 5142 | 0 7904 | 0 4816 |
| Recall of $+1/2$ defects | 0 9037 | 0 9366 | 0 8317 | 0 7461 |
| Recall of $+1/2$ defects | 0 7724 | 0 8533 | 0 5533 | 0 6095 |
| Recall Overall | 0 8649 | 0 9145 | 0 7404 | 0 6888 |
| Fscore of $+1/2$ defects | **0.7323** | 0 7392 | 0 855 | 0 6346 |
| Fscore of $+1/2$ defects | **0.4956** | 0 4941 | 0 5767 | 0 4799 |
| Fscore Overall | **0.6505** | 0 6583 | 0 7646 | 0 5669 |

Table 2: Detecting Results of YOLO

| YOLO | Overall | Dataset 1 | Dataset 2 | Dataset 3 |
|---|---|---|---|---|
| Precision of $+1/2$ defects | 0 7969 | 0 8222 | 0 8042 | 0 6655 |
| Precision of $+1/2$ defects | 0 3813 | 0 4368 | 0 3581 | 0 3216 |
| Precision Overall | 0 6608 | 0 7295 | 0 6095 | 0 469 |
| Recall of $+1/2$ defects | 0 7836 | 0 7971 | 0 6708 | 0 7645 |
| Recall of $+1/2$ defects | 0 412 | 0 3399 | 0 3909 | 0 635 |
| Recall Overall | 0 6334 | 0 6345 | 0 5118 | 0 6893 |
| Fscore of $+1/2$ defects | **0.7902** | 0 8095 | 0 7314 | 0 7116 |
| Fscore of $+1/2$ defects | **0.3960** | 0 3823 | 0 3738 | 0 4271 |
| Fscore Overall | **0.6568** | 0 6787 | 0 5564 | 0 5583 |

Table 1 and 2 displays the evaluation results for +1/2 defects, -1/2 defects, and the combination of two classes for both traditional method and YOLO on three testing datasets separately and altogether. The evaluation results include precision, recall, and F1-score values. Total testing dataset contains 1700 images, with 1500 from dataset 1, 100 from dataset 2, and 100 from dataset 3.

When evaluating the results, we consider a prediction to be correct if there is a defect located inside the bounding box generated. Based on this criterion, we evaluated our model with precision, recall, and F1-score. Overall, we achieved an overall F1-score of 0.6568, 0.7902 for +1/2 defects, and 0.3960 for -1/2 defects. In comparison, the traditional image processing defect detecting algorithm obtains 0.6505 for overall F1-score and 0.7323, 0.4956 for +1/2 and -1/2 defects, respectively. This shows that our method is currently performing as good as the traditional method overall. As observed, the detecting results for -1/2 defects are particularly poor. For this matter, we believe two factors explain the week detecting results of -1/2 defects:

- **-1/2 defects are more complicated in geometric configuration.** The fact that the traditional method also performs worse for -1/2 defects than for +1/2 ones also proves that -1/2 defects are harder to detect.

- **Unbalanced training dataset.** Due to the topological constraint, each image is required to have 2 more +1/2 defects than -1/2 defects. Therefore, within the 8800 images which we labeled, there are 58536 +1/2 defects and 24518 -1/2 defects.

## 4.2 DETECTION RESULTS ACROSS DIFFERENT REGIONS

As shown in Figure 6, for both methods, the detection performances become worse as it approaches the boundary of the system. We believe there are three factors causing the imbalance of detecting results in our method:

- **Defects appear more frequently at the central regions than near the boundary.** In the testing dataset, there are 1734, 3383, and 2750 +1/2 and -1/2 defects located at outer, middle, inner regions respectively. Since the outer region is the largest in area but contains the least number of defects, the lower detection performance is to be expected.

- **The nucleation and annihilation happens most often near the boundary.** A nucleation or annihilation process is gradual, usually taking a few frames to transition from defect to non-defect or the other way around. In these frames, it is hard even for researchers to define the boundary between defect and non-defect.

- **Defects are smaller in size near the boundary and have more various configurations.** Defects in the central region are usually well-defined and clear while the defects near the boundary are usually small in size and vague in definition. Figure 7 shows the examples of +1/2 defects located near the boundary.

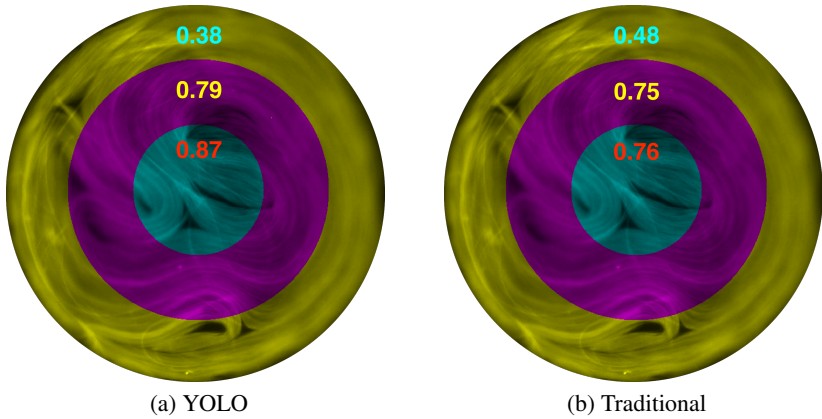

(a) YOLO          (b) Traditional

Figure 6: F1-Score Over Different Region: the detecting results evaluated on 1500 images from testing dataset. The three numbers represent the overall F1-score of all the detecting results made located within the marked region. The boundaries are set to be the 1/3 and 2/3 of the radius.

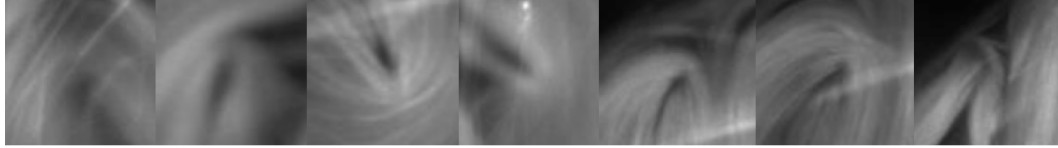

Figure 7: Examples of +1/2 Defects Near the Boundary

## 4.3 DETECTION RATE

Table 3: Detection Rates Comparison

| Method | Detection Rate(fps) |
| --- | --- |
| YOLO | 90.72 |
| Trational | 0.053 |

We evaluated the processing rate of YOLO and traditional method with 1500 testing images. Table 3 shows the detection rates for the two methods. The traditional method is executed with a Desktop equipped with RAM: 16GB; CPU: Intel Core i5-2400 @ 3.1 Ghz; GPU: NVIDEA GeForce GTS 450. We perform our data processing, model training, and detection on a machine equipped with RAM: 128GB; CPU: 2×Intel(R) Xeon(R) CPU E5-2637 v3 @ 3.50GHz; GPU: NVIDIA Corporation GP102 [TITAN X]. Although our method is operated by a machine with much stronger computational ability, our method's significant advantage in speed is still obvious.

## 5   ALGORITHM DISCUSSION

While YOLO is the state-of-the-art object detection algorithms in images, it is not designed specifically for video. In other words, YOLO is not able to make detection decisions based on information from previous and next frames. One set of data in the active nematics experiments is usually a video containing up to twenty thousand frames. Typically, defects positions do not drastically change among consecutive frames, and the active system as a whole move relatively smoothly. Therefore, improving the design of YOLO to utilize the connection between frames will be expected to improve the detecting results. This intuition is supported by research carried out by Kang et al. (2017), who developed "tubelets" with R-CNN to incorporate temporal and contextual information into the decision making process. The method, named T-CNN is a successful design which has boosted the detecting results generated under R-CNN framework. Therefore, as our next step, we will incorporate the temporal and contextual information into YOLO's pipeline in a similar way in order to improve the detecting results in our dataset.

## 6   CONCLUSION

Based on experimental results, our method has been shown to be the state-of-the-art in defects detection algorithms. Our method's advantages include significantly faster processing rate, higher F1-score on detecting results, and straightforward execution that could be operated by any researchers with simple python skills.

We have shown the viability of deep learning's application in soft matter. Most of experiments in the field of soft matter involve experimental data in the form of images or videos and object detection is one of the most common tasks physicists face. We hope this paper could provide an effective alternative for physicists when they try to tackle similar tasks.

### ACKNOWLEDGMENTS

Development of this defect detector was supported in part by the Brandeis MRSEC through grant NSF-MRSEC-1420382.

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
