# OpenReview forum: "Detecting Topological Defects in 2D Active Nematics Using Convolutional Neural Networks"
_ICLR.cc/2019/Conference_

### Official Review · AnonReviewer1 · 2018-11-02
**Application of YOLO to images of specific types**

**Rating:** 2
**Confidence:** 5

**Review:**

This review will unfortunately be very short because I am afraid there is not much to say about this well written paper, which seems to have been sent to the wrong conference. The scientific problem is interesting, namely the detection of topological artifacts in images showing biological phenomena (which I don’t know much about). The relevant literature here is basically literature from this field, which is not machine learning and not even image processing. The contribution of the paper, in terms of machine learning, is to apply a well known neural model (YOLO) to detect bounding boxes of objects in images, which are very specific. The contribution here does not lie in machine learning, I am afraid.

This is thus a purely experimental paper on a single application, namely object detection in specific images. Unfortunately the experiments are not convincing. The results are validated against a “traditional method”, which has never been cited, so we do not know what it is.

The performance gain obtained with YOLO seems to be minor, although the difference in time complexity is quite enormous (to the advantage of YOLO).

The contribution is thus minor and for me does not justify publication at ICLR.

The grant number is mentioned in the acknowledgments, which seems to violate double blind policy.

---

### Official Review · AnonReviewer3 · 2018-11-05
**A nice application of machine learning without much insight to machine learning practitioners.**

**Rating:** 4
**Confidence:** 4

**Review:**

In this paper the authors apply methods developed in computer vision towards the identification of topological defects in nematic liquid crystals. Typically, defects are identified using a costly algorithm that is based on numerically computing the winding number at different locations in the image to identify defects. The authors demonstrate that a deep learning approach offers improvement to both the identification accuracy and rate at which defects can be identified. Finally, the authors do some work investigating the limitations of the model and show that breakdown occurs near the edge of the field of view of the microscope. They show that this also happens with a conventional approach.

Overall, this seemed like a nice application of machine learning to a subject that has received significant attention from soft matter community. The results appear to be carefully presented and the analysis seems solid. However, it does not seem to me that ICLR is a particularly appropriate venue for this work and it is unclear exactly what this paper adds to a discussion on machine learning. While there is nothing wrong with taking an existing architecture (YOLO) and showing that it can successfully be applied to another domain, it does limit the machine learning novelty. It also does not seem as though the authors pushed particularly hard in this direction. I would have been interested, for example, in seeing some analysis of the features learned by the architecture trained to classify defects appropriately.

I would encourage the authors to either submit this work to a journal closer to soft matter or to do some work to determine what insights and lessons might help machine learning researchers working on other applied projects. The closest I got from the paper was the discussion of bounding box sizes and subclassification in section 3. It would have been nice to see some work discussing the dependence on this choice and what physical insights one might be able to glean from it.

---

### Official Review · AnonReviewer2 · 2018-11-06
**An application of YOLO to detect topological defects in 2D active nematics.**

**Rating:** 4
**Confidence:** 4

**Review:**

Summary:
This paper applies deep learning model YOLO to detect topological defects in 2D active nematics. Experimental results show that YOLO is robust and accurate, which outperforms traditional state-of-the-art defect detection methods significantly.

Pros:
+ Detecting defects in 2D active nematics is an important task to study.
+ YOLO is effective in object detection and shows good results for defect detection.
+ The experiment shows that YOLO appears to outperform traditional state-of-the-art defect detection methods.

Cons:
-	The technical contribution seems not enough. YOLO is state-of-the-art object detection method and has been widely used. However, this paper directly applies YOLO for this task, while few variants have been specifically designed or modified for the defect detection tasks.
-	The experiments may miss some details. For example, what is the traditional method used for comparison? What is the difference between traditional method and YOLO? The paper should provide some explanations and introductions.
-	Since the training data set is imbalanced, does the proposed model utilize some strategy to overcome this problem?
-	The detection rate comparison is not convincing. As shown in the experiments, traditional model and YOLO is operated by different machines, therefore, the detection rate comparison is not convincing.
-	The paper contains some minors. For example, in table 1 and table 2, +1/2 defects should be -1/2.

---

### Meta-Review · Area_Chair1 · 2018-12-20

**Confidence:** 5
**Recommendation:** Reject

**Metareview:**

The reviewers raised a number of major concerns including the incremental novelty of the proposed (if any) and insufficient and unconvincing experimental evaluation presented. The authors did not provide any rebuttal. Hence, I cannot suggest this paper for presentation at ICLR.